# A pilot study of the Earable device to measure facial muscle and eye movement tasks among healthy volunteers

Matthew F. Wipperman[1,2]☯*, Galen Pogoncheff[3]☯, Katrina F. Mateo[4], Xuefang Wu[4], Yiziying Chen[5], Oren Levy[2], Andreja Avbersek[2], Robin R. Deterding[3], Sara C. Hamon[1,2], Tam Vu[3], Rinol Alaj[4]*, Olivier Harari[2]*

1 Precision Medicine, Regeneron Pharmaceuticals Inc, Tarrytown, New York, United States of America, 2 Early Clinical Development & Experimental Sciences, Regeneron Pharmaceuticals Inc, Tarrytown, New York, United States of America, 3 Earable Inc., Boulder, Colorado, United States of America, 4 Clinical Outcomes Assessment and Patient Innovation, Global Clinical Trial Services, Regeneron Pharmaceuticals Inc, Tarrytown, New York, United States of America, 5 Biostatistics and Data Management, Regeneron Pharmaceuticals Inc, Tarrytown, New York, United States of America

☯ These authors contributed equally to this work.
* Matthew.Wipperman@regeneron.com (MFW); Rinol.Alaj@regeneron.com (RA); Olivier.Harari@regeneron.com (OH)

**Data Availability Statement:** All data and computer code are publicly available here: https://github.com/Earable-ML/Pilot-Study-facial-muscle-and-eye-movement-classification/tree/main/data.

## Abstract

The Earable device is a behind-the-ear wearable originally developed to measure cognitive function. Since Earable measures electroencephalography (EEG), electromyography (EMG), and electrooculography (EOG), it may also have the potential to objectively quantify facial muscle and eye movement activities relevant in the assessment of neuromuscular disorders. As an initial step to developing a digital assessment in neuromuscular disorders, a pilot study was conducted to determine whether the Earable device could be utilized to objectively measure facial muscle and eye movements intended to be representative of Performance Outcome Assessments, (PerfOs) with tasks designed to model clinical PerfOs, referred to as mock-PerfO activities. The specific aims of this study were: To determine whether the Earable raw EMG, EOG, and EEG signals could be processed to extract features describing these waveforms; To determine Earable feature data quality, test re-test reliability, and statistical properties; To determine whether features derived from Earable could be used to determine the difference between various facial muscle and eye movement activities; and, To determine what features and feature types are important for mock-PerfO activity level classification. A total of N = 10 healthy volunteers participated in the study. Each study participant performed 16 mock-PerfOs activities, including talking, chewing, swallowing, eye closure, gazing in different directions, puffing cheeks, chewing an apple, and making various facial expressions. Each activity was repeated four times in the morning and four times at night. A total of 161 summary features were extracted from the EEG, EMG, and EOG bio-sensor data. Feature vectors were used as input to machine learning models to classify the mock-PerfO activities, and model performance was evaluated on a held-out test set. Additionally, a convolutional neural network (CNN) was used to classify low-level representations of the raw bio-sensor data for each task, and model performance was correspondingly evaluated and compared directly to feature classification performance.

**Funding:** The sole funder of this study was Regeneron Pharmaceuticals, Inc. The study was designed by employees of Regeneron Pharmaceuticals, Inc, and data from this study were analyzed by employees of both Regeneron Pharmaceuticals, Inc and Earable, Inc.

**Competing interests:** The authors have read the journal's policy and the authors of this manuscript have the following competing interests: MFW, KFM, XW, YC, OL, AA, SCH, RA, and OH are current or former employees and shareholders of Regeneron Pharmaceuticals, Inc. GP, RRD, and TV are employees of Earable, Inc.

The model's prediction accuracy on the Earable device's classification ability was quantitatively assessed. Study results indicate that Earable can potentially quantify different aspects of facial and eye movements and may be used to differentiate mock-PerfO activities. Specially, Earable was found to differentiate talking, chewing, and swallowing tasks from other tasks with observed F1 scores >0.9. While EMG features contribute to classification accuracy for all tasks, EOG features are important for classifying gaze tasks. Finally, we found that analysis with summary features outperformed a CNN for activity classification. We believe Earable may be used to measure cranial muscle activity relevant for neuromuscular disorder assessment. Classification performance of mock-PerfO activities with summary features enables a strategy for detecting disease-specific signals relative to controls, as well as the monitoring of intra-subject treatment responses. Further testing is needed to evaluate the Earable device in clinical populations and clinical development settings.

## Author summary

Many neuromuscular disorders impair function of cranial nerve enervated muscles. Clinical assessment of cranial muscle function has several limitations. Clinician rating of symptoms suffers from inter-rater variation, qualitative or semi-quantitative scoring, and limited ability to capture infrequent or fluctuating symptoms. Patient-reported outcomes are limited by recall bias and poor precision. Current tools to measure orofacial and oculomotor function are cumbersome, difficult to implement, and non-portable. Here, we show how Earable, a wearable device, can discriminate certain cranial muscle activities such as chewing, talking, and swallowing. We demonstrate using data from a pilot study how Earable can be used to measure features from EMG, EEG, and EOG waveforms from subjects wearing the device while performing mock Performance Outcome Assessments (PerfOs), utilized widely in clinical research. Our analysis pipeline provides a framework for how to computationally process and statistically rank features from the Earable device. Our results, conducted in a pilot study of healthy participants, enable a more comprehensive strategy for the design, development, and analysis of wearable sensor data for investigating clinical populations. Understanding how to derive clinically meaningful quantitative metrics from wearable sensor devices is required for the development of novel digital endpoints, a hallmark goal of clinical research.

## Introduction

Facial/cranial and eye movement dysfunction is an important feature of several neurological disorders that affect multiple levels of the neuraxis. [1] Examples include outright facial weakness due to facial nerve palsy or stroke, diplopia, ptosis, and dysphagia caused by neuromuscular disorders such as myasthenia gravis, dystonia, complex extraocular movement deficits, hypomimia, and dysphagia caused by parkinsonian (and other neurodegenerative) conditions. [2,3]

Clinical assessment of these symptoms remains a challenge in medicine and clinical research. [4,5] Existing clinical assessments, such as clinician-reported outcomes (ClinROs) or patient-reported outcomes (PROs) may require the patient to frequently visit sites, primarily rely on subjective measures, and may not necessarily reflect a patient's condition(s) in the real

world. Importantly, patient symptoms can be intermittent and vary throughout the day, making reliable assessment difficult. Finally, they can be variable from patient-to-patient depending on their adaptations to increasing muscle weakness.

While there are tools that exist to perform quantitative analysis of cranial muscle function, these tools have significant limitations. For example, facial movements can be measured with video-based technologies using either static images or video capture. [6] Surface Electromyography (EMG), which records the electrical movements of facial muscles, can also be used either alone or in combination with video-based methods. [7] Small studies have suggested that Electrooculography (EOG), which measures electrical potential from the front to the back of the eye, can detect differences between parkinsonian patients and controls. [8,9] Screen-based trackers and wearable glasses have been used to monitor extraocular movements and upper cranial activity (e.g. blinking). [10] Together in their current application, these approaches can be cumbersome, difficult to implement, and most importantly, capture facial movements for brief periods of time in an artificial setting.

As such, there is an opportunity to identify and/or develop novel non-invasive approaches to measure cranial symptoms of neuromuscular and neurodegenerative disorders to address these problems in key patient populations. If properly studied and validated, these tools may in turn serve to support diagnostic and disease progression assessments by clinicians, but also outcomes assessment in clinical research. If such approaches can leverage wearable sensing technology, they may be able to address the challenges of existing clinical sensors that are limited for use in highly controlled settings, as opposed to more naturalistic environments (e.g., at home).

One such wearable device, Earable is a behind-the ear device developed to measure neural and physiological processes. [11] Electrophysiological signals are acquired at 250 Hz via four re-usable electrodes fabricated from a conductive silicon material. Electrodes of the device are positioned at scalp locations directly above the left and right ears and on left and right mastoid processes, yielding raw bio-signal data analogous to that which could be acquired at Electroencephalography (EEG) reference locations T3, T4, M1, and M2 of the 10–20 electrode placement positions [12], EEG being a measurement of surface brain wave function. This electrode configuration also enables high fidelity acquisition of EMG activity from activation of the temporalis and surrounding muscle groups, and EOG signals yielded by eye deflections.

Whereas traditional clinical assessment using biophysiological data may be invasive, expensive, and time consuming, Earable is purposed to offer high fidelity data acquisition and processing to the general population. The EMG, EEG, and EOG signals monitored with Earable have been used for the detection and evaluation of a wide variety of physiological phenomena, such as sleep monitoring [13], microsleep detection [14], and acute postoperative pain quantification. [15] Earable has the potential to support outcome assessment for neuromuscular disorders by objectively quantifying facial muscle and eye movement tasks through capturing and analyzing bio-signal data. A significant challenge is that unprocessed bio-signal data is inherently noisy due to several factors, e.g. participants move during clinical assessments, there may be perturbations in electrode-skin contact, there are artifacts from cardiac activity. Additionally, similar factors naturally induce artifacts in the acquired signal data; EEG, EMG, and EOG signals overlap in typical frequency ranges, making direct separation and analysis of the waveform data non-trivial.

Thus, as an initial step to develop a digital assessment for neuromuscular disorders, a pilot study was conducted to determine whether the Earable device could measure facial muscle and eye movements. The specific aims of the study were:

- To determine how the Earable EMG/EOG/EEG signals may be processed to extract features;

- To determine Earable feature data quality, test re-test reliability, and statistical properties;

- To determine whether parameters derived from the Earable device can quantify various facial and ocular muscle activities;

- To determine what features are important for activity level classification, in comparison to raw bio-signal data classification approaches.

In this pilot study, we developed 16 mock Performance Outcome Assessments (mock-PerfOs) designed to assess facial and eye movements with the Earable device on N = 10 control volunteer participants. [16] We present our approach of a fit-for-purpose feature engineering pipeline, where we derive features from the EMG, EOG, and EEG waveforms, evaluate feature relationships to each other, and quantitatively assess how features classify different mock-PerfO activities. The steps taken in this study reflect the analytical validation steps of the V3 framework for the development of digital assessments. [17] Taken together, the results from our study highlight the utility of the Earable device and similar devices that collect bio-signal data as potential measurement tools in a clinical trial setting for evaluating facial and eye movement tasks, and enable further clinical development with this and similar devices.

## Results

### A pilot study was run to evaluate how Earable can distinguish between facial muscle and eye movement activities

To evaluate how well Earable can classify facial muscle and eye movement tasks, we conducted a pilot study with N = 10 participants who performed 16 facial muscle task movements (mock-PerfOs) four times in the morning and four times in the evening. Table 1 shows the demographic characteristics of the study participants.

Earable raw bio-signal data was processed into 161 summary features, most of which describe the EMG, EOG, and EEG waveforms. The process to summarize the raw Earable bio-signal data into features is described in detail in the Methods. Briefly, features were computed from EMG, EOG, and EEG waveform components that were separated from raw, mixed-waveform bio-signals through specialized signal combination and filtering mechanisms. A high-level overview of the feature engineering process is summarized in Fig 1. After processing each waveform into its component parts, an event-based segmentation algorithm was applied, and feature computation was performed (Fig 1A). Features are representative of both the frequency and time domain of the Earable signal, illustratively shown in Fig 1B for a subject drinking water. Qualitative differences can be observed from representative signals from the 16 mock-PerfO activities (Fig 1C).

At a high level, features can be described in categories described in Supplemental Table 1. Amplitude features, zero crossing rate, standard deviation, variance, root mean square

**Table 1. Study sample characteristics.**

| Characteristic | Mean (SD) |
|---|---|
| **Sex** | |
| Male (N, %) | 5, 50% |
| Female (N, %) | 5, 50% |
| **Age** (years) | 32.03 (11.66) |
| **Height** (inches) | 71.33 (14.83) |
| **Weight** (pounds) | 154.8 (21.83) |
| **BMI** (kg/m$^2$) | 22.91 (5.33) |

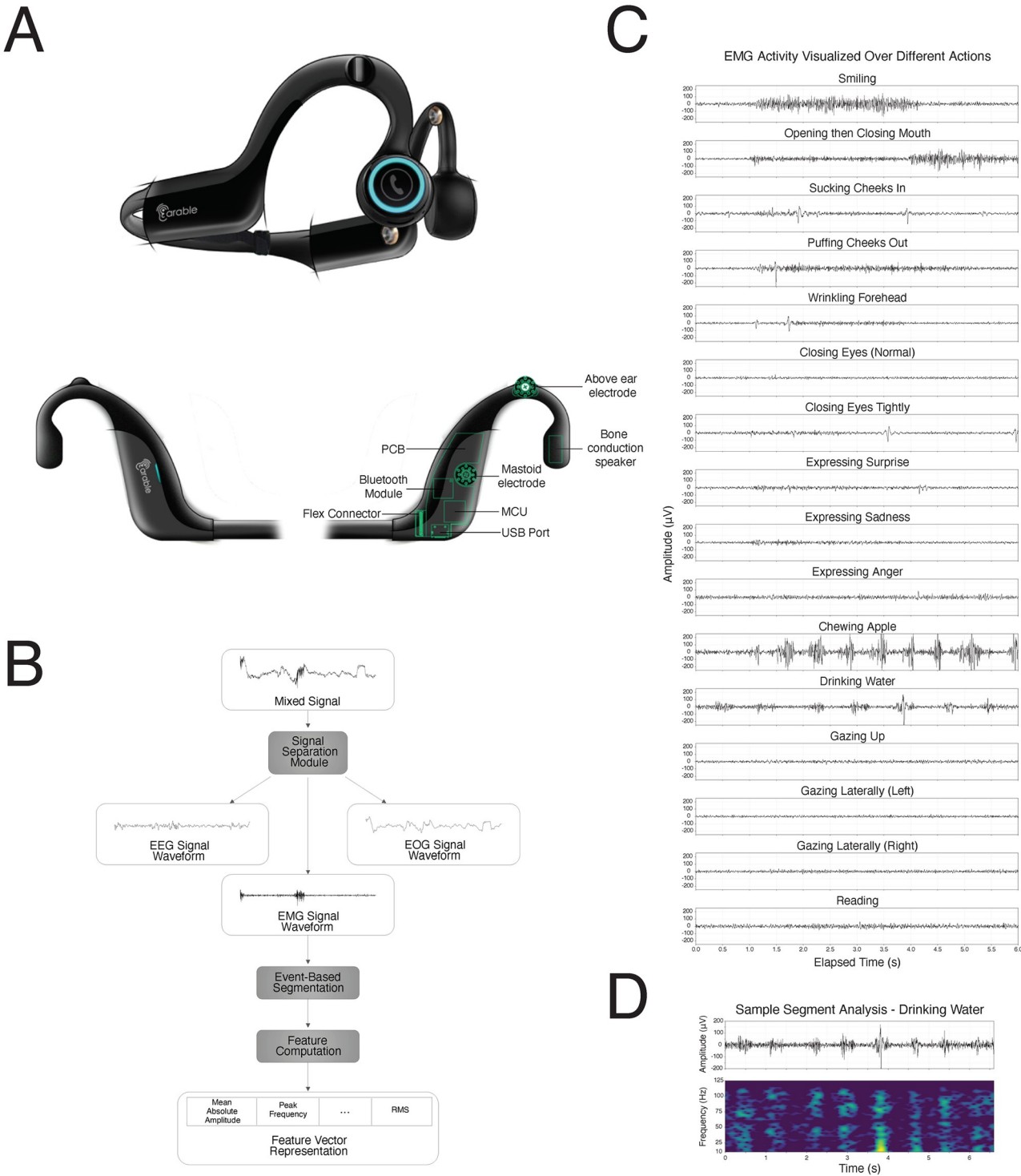

**Fig 1. A.** A schematic of the Earable device used in this pilot study. The left and right earpiece components, which sit over the ears of the participant, house the primary electrical components of the device and are connected by a flexible circuit within a connecting band that allows the device to be worn comfortably by patients with varying head shapes and sizes. The microcontroller unit (MCU) enables data acquisition, signal processing, and on-device data processing. A Bluetooth module is incorporated for data streaming directly to a local computer or mobile phone. Multi-channel, electrophysiological data is acquired through dry electrodes that contact the participant's scalp above each over and over each mastoid process. Although not directly applicable to this pilot study, music and audio-based stimulation and therapy is made available to the patient via bone-conduction speakers. **B.** Signal Processing and feature extraction pipeline used in this study. A signal separation module is applied to the mixed signal derived from the Earable device, to separate the EEG, EMG, and EOG waves into their component parts. These signals are then subject to an event-

based segmentation algorithm, and features extracted. **C.** Time and frequency representations of EMG activity resulting from a participant drinking water. This plot shows around 6.5s of EMG data in both the time (top) and frequency (bottom) domains. **D.** EMG activity visualized in the time domain over 16 activities.

[18–22], kurtosis [23,24], frequency, bandpower [25], skew [26], as well as other standard waveform features were processed from Earable sensor data. Features were selected according to standard feature processing pipelines. Amplitude features describe the amplitude or maximum distance from baseline of each wave in the relevant component space. Bandpower features describe the average power of a wave in a specific frequency range (where there are multiple frequency ranges specific to each signal type for Earable). Other named features mathematically describe the shape, variance, or complexity of the EMG, EOG, or EEG waveforms.

## Earable parameters measure unique aspects of facial and eye movement

We investigated the relationship between the parameters across all 16 mock-PerfO activities by performing Spearman correlations of all parameters against each other. To determine the optimal number of parameter groups that describe the variation in the overall signal, we performed k means clustering of the Spearman correlations of all parameters for all activities against each other and determined 6 unique clusters of parameters (Fig 2A). Amplitude and Bandpower parameters tended to cluster together in two of the six clusters, while other parameters like those from the frequency domain clustered separately.

To investigate qualitative differences between the 16 mock-PerfO activities (across each participant and timepoint, and all 161 Earable parameters), we performed Uniform Manifold Approximation and Projection (UMAP) dimensionality reduction and observed qualitative differences between the 16 mock-PerfO activities (Fig 2B). While there was overlap between some of the activities, activities like swallowing clearly separated out from the rest with this approach. We also constructed heatmaps of parameter z-score across all data that demonstrate differences between the activities for different classes of parameters (Fig 2C). Taken together, these results demonstrate the utility of the Earable device to generate parameters that may describe unique mock-PerfO activities.

## Earable has favorable feature test re-test reliability

To evaluate Earable feature test re-test reliability, we used linear mixed effects modeling with participants as random effects to evaluate feature properties (S1 and S2 Tables). First, for each of the 161 Earable features, we computed the intraclass correlation coefficient (ICC) for participants to assess the variance associated with each person for each feature (S3A Table). The ICC is a measurement of how similar and thus reliable the same data from the same participant are for the same activity, and ranges from 0 to 1 (ICC less than 0.5 would be poor reliability, an ICC of 0.5–0.7 would be moderate reliability, and ICC greater than 0.7 may be interpreted as a reliable metric). ICC values ranged from 0–0.92, and the average ICC value for all parameters across the 16 activities was 0.31. Second, we computed Coefficients of Variation (%CVs) for each parameter within a participant across timepoints (morning and evening) (S3B Table). Third, we computed descriptive statistics for each feature (S3C Table). Finally, we computed the variance for each feature for each activity, associated with time-of-day activities were performed (morning or evening), individual participants themselves, and individual trial repeats, as well as the unexplained variance (S3D Table). Taken together, these results support many Earable parameters as reliably measuring intra-participant variation and provide a metric by

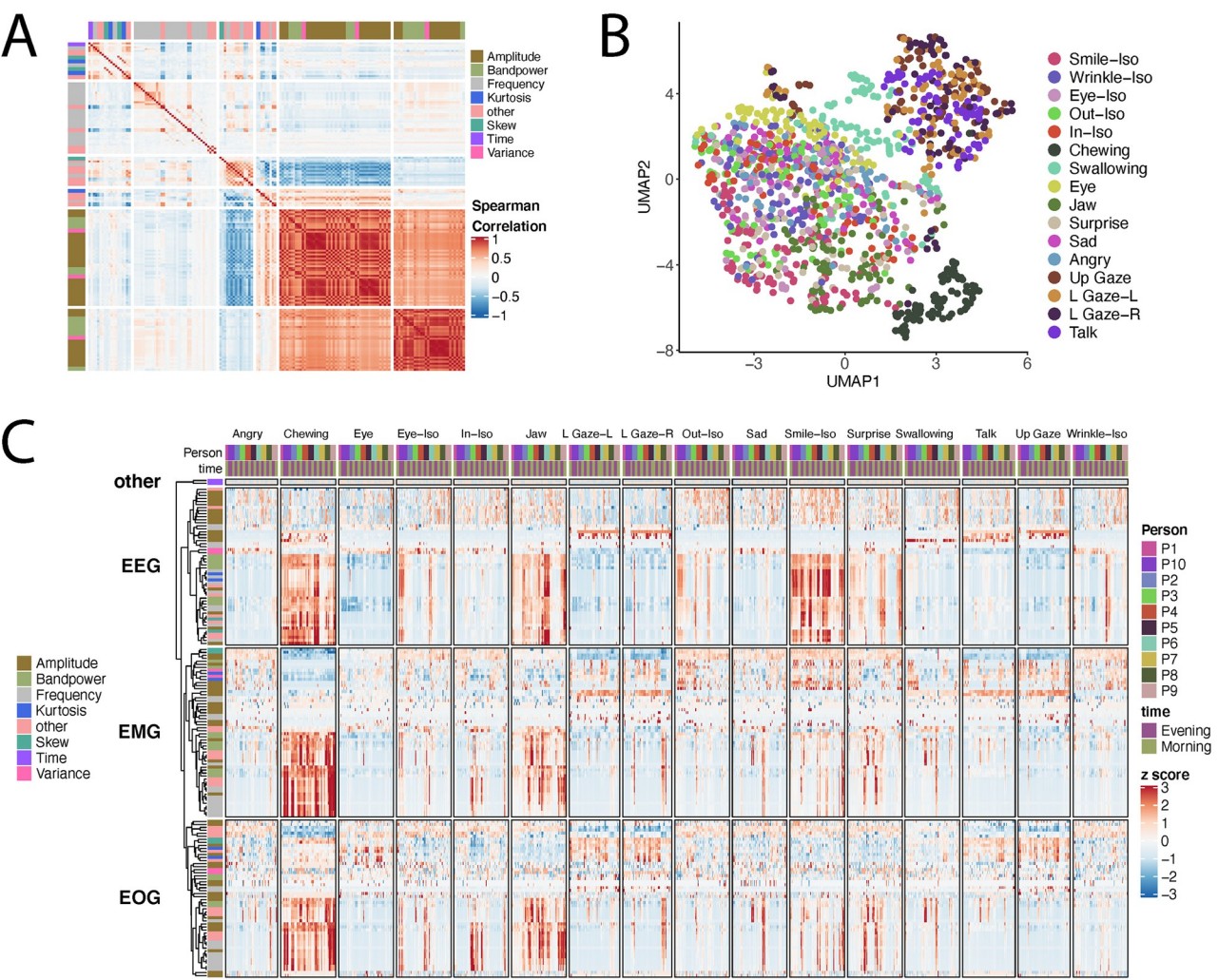

**Fig 2. Dimensionality reduction and visualization of Earable features. A.** Spearman correlation of all 161 Earable features against each other, represented as a heatmap. K = 6 clusters from K means clustering (optimal number) are shown. All 16 mock-PerfOs were pooled for the correlation analysis. **B.** UMAP dimension reduction of all 161 Earable features. Each individual activity repeat is a point on this graph. The color of the point represents the activities performed during that activity. **C.** Heatmap of all 161 Earable features (rows) for all activity repeats (columns). Columns are sorted first by the 16 activates in the pilot study, and within each activity, by participant, and then time of day when the activity was performed.

which one may rank candidate features for further downstream analysis. Further, the ICC provides a measure of internal consistency for each activity, given our expectation that subjects should generate similar outputs for the same activity.

## Earable can accurately classify some facial muscle movement activities

To investigate whether the Earable data was able to classify any of the 16 mock-PerfO activities, a Random Forest classification model was constructed to detect each activity from the other 15 activities (1-against-all classification). Activity detection F1 scores were used as the primary metric for evaluating model performance (see Methods).

Following developmental evaluation on the testing dataset for all 161 features, a second model was built for activity-level classification. This second model used an optimized set of Earable features with the goal of eliminating noisy features that would not contribute to overall

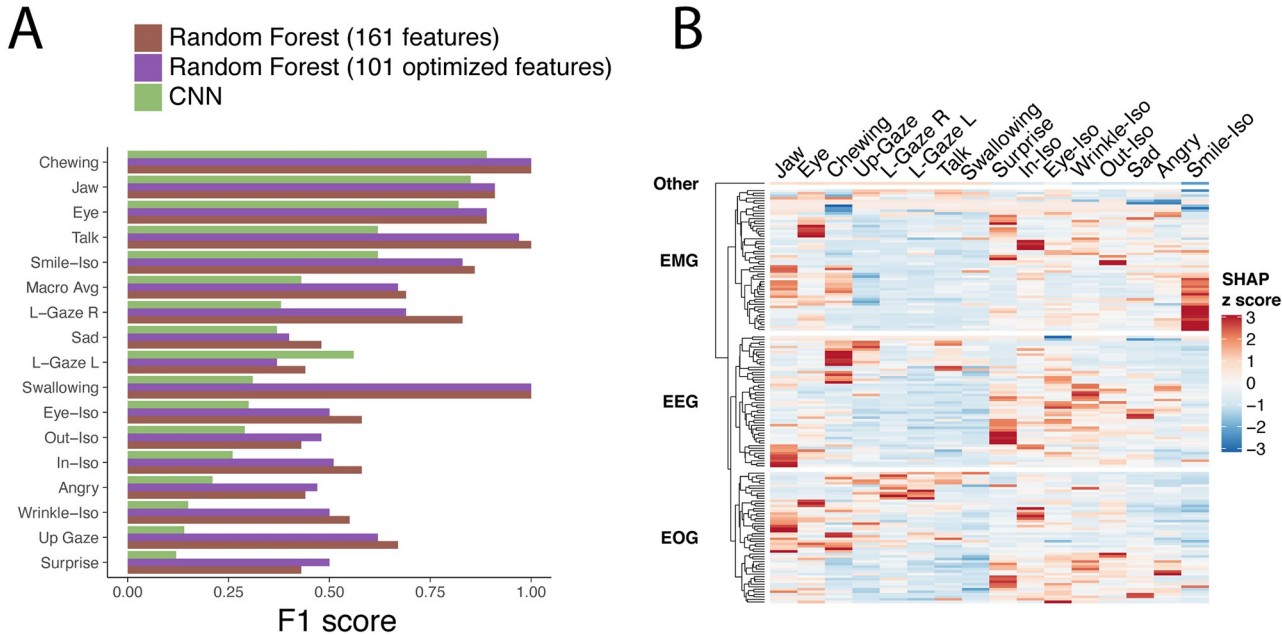

**Fig 3. A.** Activity level classification F1 scores for all Earable features (161 features), Boruta selected Earable features (101 features), and using raw waveform data (CNN). F1 scores range from 0 to 1, with 1 indicating perfect classification. **B.** Feature attribution analysis using SHapley Additive exPlanations (SHAP) values for each feature (row) for each activity (columns) determined on the model from the full set of 161 features. [28] SHAP values were z scored across all activities.

classification performance. For determination of the optimized set of Earable features, we performed feature reduction with the *Boruta* package (see Methods). [27] Briefly, all 161 Earable features are copied (called shadow features), and their class labels are randomly shuffled. Each shadow feature is compared to the real values for 1,000 iterations of classification, and only features that perform better than chance are kept. This analysis indicated an 'optimized' set of 101 Earable features that were be used for a second classification model.

Finally, to evaluate how well Earable features perform relative to using low level representations of the Earable waveform data, we built CNN models with the Earable raw bio-signal data to classify the 16 mock-PerfO activities (S1 Fig). In this modeling, fixed size spectrograms that quantify how the power at a given frequency changes as a function of time were computed from the mock-PerfO signal segments and used as model input.

As shown in Fig 3A, we compared the F1 scores for classification accuracy of the full set of 161 features, the optimized set of 101 features, as well as the predictions from the CNN (Fig 3A). To determine the underlying features that are most important for the full RF model with 161 features, we used feature attribution analysis with SHapley Additive exPlanations (SHAP). [28] For a specific activity prediction, the SHAP value of an Earable feature was computed as the change in the expected value of the model output when this feature is observed, compared to when it is missing for the test set predictions. The effect of each feature as it is added to the model is summed and averaged across all 161 Earable parameters used. The parameters are represented as the Mean average SHAP value and are shown in a heatmap (log10) (Fig 3B). Finally, we determined the percent contribution for each activity of each waveform group of features (Table 2). The normalized sum of absolute SHAP values for each activity was compared against the sum within the EMG, EEG, and EOG features, and normalized by the number of features in that group to calculate the percent contribution of each waveform to the classification accuracy.

**Table 2. Table of the 16 mock-PerfO activities and how EMG, EEG, and EOG feature groups contribute to classification accuracy.**

| Activity | Normalized Sum of Absolute SHAP Values | EMG % Contribution | EEG % Contribution | EOG % Contribution |
|---|---|---|---|---|
| L Gaze-R | 0.517996133 | 48.5 | 21.5 | 30.0 |
| L Gaze-L | 0.49281577 | 48.1 | 22.9 | 29.0 |
| Up Gaze | 0.526086224 | 48.2 | 27.9 | 23.9 |
| Chewing | 0.669707634 | 49.8 | 27.3 | 22.9 |
| Eye-Iso | 0.50169248 | 58.1 | 22.3 | 19.5 |
| Out-Iso | 0.478709587 | 58.3 | 22.6 | 19.1 |
| Eye | 0.671975561 | 62.9 | 18.1 | 18.9 |
| In-Iso | 0.553403349 | 60.4 | 20.8 | 18.8 |
| Jaw | 0.759305572 | 62.4 | 20.5 | 17.1 |
| Surprise | 0.502448985 | 57.5 | 25.7 | 16.8 |
| Wrinkle-Iso | 0.528007475 | 56.7 | 26.7 | 16.6 |
| Talk | 0.577886127 | 61.4 | 22.4 | 16.2 |
| Angry | 0.485578854 | 66.6 | 18.0 | 15.4 |
| Sad | 0.468684661 | 67.4 | 17.4 | 15.3 |
| Swallowing | 0.784064192 | 68.9 | 17.9 | 13.2 |
| Smile-Iso | 0.558172499 | 75.6 | 16.0 | 8.4 |

Shown are the Normalized sum of the Absolute SHAP values from the RF model, as well as the relative EMG, EEG, and EOG percent contributions to classification importance. Feature importance was normalized based on the total number of features in each EMG, EEG, or EOG group, compared to the total number of features in all three categories. Features not associated with any waveform were excluded from this analysis.

## Discussion

Improving pipelines for the development and analysis of wearable sensor data and frameworks for how to think about these data in clinical settings is critical for improving accurate patient diagnosis and monitoring treatment responses in all stages of clinical drug development. [17,29] Despite the progress made to date, there still exists challenges in both the development of wearable devices themselves as well the ways in which wearable data are processed and analyzed in clinical settings. [17,30]

In this work, we demonstrate a proof-of-concept repurposing of a wearable device, Earable (a sleep aid wearable that measures EMG, EEG, and EOG), to assess facial and ocular muscle movements in a pilot study of healthy controls. We highlight the utility of a feature engineering approach to classify activities intended to be representative of true PerfOs. Further, we present our approach for analyzing and ranking the utility of features generated from Earable and discuss how this data may be used to classify activities performed in this study setting. Finally, we highlight limitations of time series analysis on bio-signal data collected over short periods of time compared to a feature-based analytical approach, something we feel is an important consideration for wearable data analysis pipelines.

We demonstrated that data generated by Earable can be used to classify certain types of cranial muscle and ocular movements. The data generated in this pilot study suggest that while further work is needed to refine the types of activities more accurately to be employed as PerfOs in a clinical setting, Earable could potentially be used to objectively monitor certain cranial movements, such as eye blinking rate, which is increased in some neuromuscular disorders such as ocular myasthenia gravis and reduced in parkinsonian disorders. [31,32] Additionally, there may be unrealized advantages of measuring multiple types of waveforms simultaneously

from a single device, given the demonstrated utility of these waveforms to measure disease in clinical settings.

Interestingly, data from this study were consistent with our expectations of which activities may relate to which types of waveforms. For example, feature importance analyses indicated that EOG was associated with contributing largely to gaze or eye movement activities (up, left, and right) when analyzing which features were most important at classifying activities using the full RF model (Table 2). These types of activities would be expected to have EOG as a significant contributing factor, and while we saw that the EMG signal is overall most important for activity classification, the other components do play an important role. In a limited number of cases, the presence of signal artifacts was observed to obfuscate waveform contribution analysis. For instance, this was notable in the Chewing activity, where residual EMG activity that overlapped with typical EEG frequencies persisted in the EEG signal after signal separation, resulting in an overestimate of EEG waveform contribution.

While further research and additional clinical validation data are necessary, we feel that there are numerous neuromuscular and/or neurodegenerative conditions that may benefit from improved use of wearable sensor technology like Earable. The main goal of a feature extraction pipeline from specific waveforms as described in this work is to support the development of novel digital endpoints for use in clinical trial settings. These features, either alone or in combination with other features or other types of data generated in a trial, may form the basis for future clinical endpoints after further evaluation for how they measure disease progression or treatment response.

Challenges in this pilot study included several feature engineering and evaluation considerations. We chose to directly compare classification accuracy (F1 scores) of models built from both processed sensor data, as well as from raw bio-signal data. Interestingly, we found that regardless of data augmentation, regularization, and other techniques used to counter overfitting (see Methods), the training dataset was observed to be too small to train a generalizable CNN model. However, in clinical settings, the level or amount of data collected in this pilot study may in fact be representative of data collected in a clinical laboratory setting. As such, understanding the most appropriate analysis method for a particular clinical question is of great utility and importance. We note that different analytical approaches may be more suitable for certain types of questions, and there is no "one size fits all" datatype or model that can address all questions. Our findings suggest that there may be some limitations of time series models applied to bio-signal collection data common in clinical research, especially brief (seconds to minutes) PerfO activities. [16]

Limitations of this study include the small sample size, especially with respect to more generalizable claims about device usability (in a real-world setting). Additionally, this study was run in healthy control participants, and thus there is difficulty in extrapolating results to a disease population. Future assessments for verification and analytical validation [17] will include: 1) the testing of the device for usability in relevant patient populations, and 2) the use of the device with true PerfO activities for reference dataset creation in disease populations.

Despite the above caveats, data from this study suggest that Earable, as well as similar wearable devices, may be promising tools for further development in clinical research settings, opening the door to more objective quantitation of cranial and eye muscle movements. Future clinical validation [17] work in this space will focus on the clinical utility of the Earable analysis pipeline to: 1) test the utility of the Earable device in disease populations, 2) more accurately measure disease progression within participants, 3) test how Earable features or data relate to existing PROs, and finally 4) more accurately measure treatment effects within disease populations, hallmark goals in early clinical development. The use of Earable in longitudinal studies where disease progression may be measured, for example ongoing natural history studies, may

help elucidate which features are most important for quantifying disease effects. Finally, the exploratory use of these devices in clinical trials as part of a wearable clinical development strategy may enable more sensitive detection of treatment responses within disease populations. These clinical validation steps may additionally support a strategy to use devices like Earable for passive monitoring purposes.

## Methods

### Ethical statement and study approval

All participants provided written informed consent prior to the study.

### Study participants

A total of 10 healthy volunteers were recruited for this pilot study. All participants were screened according to the inclusion and exclusion criteria listed below.

### Inclusion and exclusion criteria

Candidates had to satisfy the following to be enrolled in the study:

- The candidate age 18 or older

- The candidate demonstrates the ability to understand the Informed Consent Form (ICF) and the willingness to follow all study instructions.

- The candidate has read and signed the ICF

The presence of any of the following eliminated the candidate from enrollment in the study:

- The candidate is pregnant

- The candidate has been diagnosed with the following conditions: muscular dystrophy, myasthenia gravis, Amyotrophic lateral sclerosis, multiple sclerosis, spinal muscular atrophy.

### Study tasks

All participants were asked to complete two 45-minute sessions. During each session, each participant was asked to complete a series of tasks listed in Table 3 below. Participants were asked to take a one-minute break between each task.

These tasks were chosen to represent tasks patients with craniofacial neuromuscular disorders may have difficulty completing. Study tasks were based on activities performed during routine neurological examination of cranial nerves, as well as grading systems for facial weakness. [33,34] These activities are commonly used to diagnose and assess severity of several diseases that affect cranial muscles, including facial nerve injury, stroke, and neuromuscular disorders such as myasthenia gravis and amyotrophic lateral sclerosis.

### Study procedure

Each study participant engaged in two study sessions, one in the morning and one at night. Testing sessions were conducted one-on-one by a study moderator. In the morning session, the study moderator reviewed the informed consent form (ICF) with the participant, ensured that he/she understood the form and agreed to participate. The participants had time to ask questions before signing the ICF.

**Table 3. Tasks and duration of each task study session.**

| Task | Duration for each task | Repeats | Total duration (including task preparation) |
|---|---|---|---|
| ICF, introduction and Q&A | 5 minutes | N/A | 5 minutes |
| Earable device setup | 5 minutes | N/A | 5 minutes |
| Smile broadly and show teeth as hard as possible | 15 seconds | 4 | 1 minute |
| Wrinkle forehead as tightly as possible | 15 seconds | 4 | 1 minute |
| Close eyes normally | 5 seconds | 4 | 20 seconds |
| Close eyes as tightly as possible | 15 seconds | 4 | 1 minute |
| Puff out cheeks as much as possible | 15 seconds | 4 | 1 minute |
| Suck in cheeks as much as possible | 15 seconds | 4 | 1 minute |
| Chewing | 30 seconds | 4 | 2 minutes |
| Swallowing | 10 seconds | 4 | 40 seconds |
| Upwards gaze | 45 seconds | 4 | 3 minutes |
| Lateral gaze-left | 45 seconds | 4 | 3 minutes |
| Lateral gaze-right | 45 seconds | 4 | 3 minutes |
| Talking | 30 seconds | 4 | 2 minutes |
| Open and close the jaw as much as possible | 15 seconds | 4 | 1 minute |
| Facial expression-surprise (for explorative purposes only) | 15 seconds | 4 | 1 minute |
| Facial expression—sad (for explorative purposes only) | 15 seconds | 4 | 1 minute |
| Facial expression—angry (for explorative purposes only) | 15 seconds | 4 | 1 minute |
| **Total Time** | | | 33 minutes |

The study moderator read a study script, which provided a study overview and description of various study activities. The study moderator then collected participants' baseline (background) information.

The study moderator then had participants perform the following at each study session:

1. Smile broadly and show teeth as hard as possible

2. 1-minute break

3. Wrinkle forehead as tightly as possible

4. 1-minute break

5. Close eyes as tightly as possible

6. 1-minute break

7. Put out cheeks as much as possible

8. 1-minute break

9. Suck in cheeks as much as possible

10. 1-minute break

11. Chewing for 30 seconds

12. 1-minute break

13. Swallowing

14. 1-minute break

15. Close eye normally for 5 seconds

16. 1-minute break

17. Talking 30 seconds

18. 1-minute break

19. Upward gaze for 45 seconds

20. 1-minute break

21. Lateral gaze left for 45 seconds

22. 1-minute break

23. Lateral gaze left for 45 seconds

24. 1-minute break

25. Open and close jaw as much as possible

26. 1-minute break

27. Facial expression—surprise

28. 1-minute break

29. Facial expression—sad

30. 1-minute break

31. Facial expression—angry

## Participants' de-identification, confidentiality and data protection

Participants in this study were de-identified. The study moderator assigned a unique code number to each participant as a means of referencing participants, such that during data analysis, study team members could not associate participants' names or any other unique personal identifiers with the study data.

The study moderator took all appropriate measures to ensure that the anonymity of each study participant would be maintained. Participants were identified by their initials and a participant identification number only.

## Data transfer, storage and consolidation

All study data was transferred to a shared data storage platform within 2 business days of data collection. Only approved study team members had access to this platform. All task data was labelled with the below annotations (Table 4).

## Earable raw sensor data processing and feature engineering

Raw Earable data was continuously collected during each activity of the pilot study. To guarantee reliable ground truth data annotations, data from each activity was manually labeled by an expert technician. For each activity, the onset and offset endpoints of each performed activity were annotated accordingly. A time-synchronized video recording of the participant was utilized as a reference source in this annotation procedure. Using these activity annotations, signals were then segmented according to noted onset and offset timestamps.

After completion of the activity, the resulting signals from each channel were scaled to counteract the effects of amplification performed in the device hardware for the purpose of

**Table 4. Task label annotation.**

| Task | Label |
|------|-------|
| Smile broadly and show teeth as hard as possible | Smile-Iso |
| Wrinkle forehead as tightly as possible | Wrinkle-Iso |
| Close eyes as tightly as possible | Eye-Iso |
| Puff out cheeks as much as possible | Out-Iso |
| Suck in cheeks as much as possible | In-Iso |
| Chewing | Chewing |
| Swallowing | Swallowing |
| Close eyes normally | Eye |
| Open and close jaw as much as possible | Jaw |
| Upwards gaze | Up Gaze |
| Lateral gaze-left | L Gaze-L |
| Lateral gaze-right | L Gaze-R |
| Talking | Talk |
| Facial expression—surprise | Surprise |
| Facial expression—sad | Sad |
| Facial expression—angry | Angry |

noise suppression and filtered offline using a second-order infinite impulse response (IIR) notch filter to remove 60 Hz power line noise. At this stage, each signal contained a mixture of EEG, EMG, and EOG data. A signal separation algorithm was applied to better isolate each of these components, yielding a total of six channels (two each for EEG, EMG, EEG).

Following signal scaling, filtering, and separation, the signals of each of the six separated channels were segmented based on the presence or absence of facial movement activity (Fig 1A). A comprehensive approach to feature extraction was taken for further downstream analysis. We chose to process most general features to summarize each waveform, apart from a subset of features specific to EMG, EOG, or EEG activity. Features that would clearly identify mock-PerfO activities performed within the data collection process but would not generalize to performance of the activity outside of laboratory contexts were omitted (for instance, duration of an activity that each participant was instructed to perform for a specified period).

Statistical measures from each separated signal segment were computed to summarize signal behavior in the time-domain. Such measures enable depiction of information such as time-varying amplitude behavior, amplitude distributions, and signal trends observable in their raw forms. As the frequency and time-frequency domains also contain vast amounts of information in bio-signal data, digital signal processing (DSP) analyses were performed to decompose each separated signal segment into frequency components and evaluate patterns in this alternative representation (Fig 1). Furthermore, handcrafted features relevant to theoretical EMG, EOG, and EEG behavior during specific mock-PerfO activities were computed to better represent such activities in the summary feature vectors.

Together, this yielded 161-dimension feature vector representations for each mock-PerfO activity performed. The features and their high-level categories are described in S1 Table. To remove features potentially irrelevant to activity-based classification, we implemented feature reduction with the Boruta package [27], yielding a lower dimensionality feature vector representations of each mock-PerfO activity. In this process, 60 features that were estimated as "unimportant" were removed from each feature vector, resulting in 101-dimension feature vectors. A Python implementation of the *Boruta* package (*BorutaPy*, version 0.3) was used to preform feature reduction.

## Statistical analyses

**Correlation of Earable parameters and differences in parameters between activities.**
Spearman correlations between all parameters and all activities were computed. We used the silhouette method to determine the optimal number of clusters with the *factoextra* package in R with function fviz_nbclust with 100 bootstrapped samples.

For each of the 16 activities, for all the 161 computed Earable parameters, we report the number of tasks analyzed (n), the minimum value (min), maximum value (max), median value (median), mean value (mean), standard deviation of the mean (sd), and standard error of the mean (se) (S3 Table).

**Relationships between Earable parameters and activity or demographic information.**
For data from the pilot study, the intraclass correlation coefficients (ICC1) for participants as the group was computed using linear mixed-effects modeling with the *lmer* package in R, with the following formula: ~(1|participant). ICC1 was computed separately for each of the 16 activities for each of the 161 Earable parameters (S3 Table). Coefficients of variation were also computed comparing within each activity (S3 Table).

We additionally computed the within and between trial variability due to repeated measures, time of day, and participants, as well as the variance not explained by these three factors (S3 Table). We used a nested linear mixed effects model to derive the variation explained by each component: ~ 1+ (1|time) + (1|participant) + (1|repeat/time), where the time component indicates time of day (morning or evening), the participant component indicates the subject, and the repeat component indicates the repeat of the same activity nested within the same time. The percent contribution of each of these variance components are reported in S3 Table.

**Earable data visualizations.** Dimensionality reduction of Earable parameters was performed in Python with umap-learn, with an effective minimum distance between embedded points of one, and default parameters. UMAP coordinates were plotted with ggplot2 in R. Heatmaps of Earable parameters are displayed with individual activities (trials) as columns and Earable parameters as rows. All heatmaps of Earable data display z-scored parameter rows, computed across all activities. Heatmaps were constructed with the *ComplexHeatmap* package in R.

**Quantifying pilot study activities and participant-level predictions.** To investigate how Earable features could be used to classify each of the 16 activities we implemented multi-class classification models using the Python sklearn module. We used a random forest classifier (using the sklearn RandomForestClassifier class) with 500 decision trees for model building. In each classification setting model training and validation was performed using 80% of the dataset while the remaining 20% of the dataset with withheld for testing. Data samples were assigned to one of the two subsets at random to reduced bias in evaluation results. The F1 score was calculated to evaluate model performances on the test set. The F1 score is the harmonic mean of precision and recall and elucidates the number of predictions that were accurate by the model, balancing both false negatives and false positives.

**CNN model of activity level prediction.** In recent years, Deep Learning models have been used to achieve high performance in many tasks relevant to classification of bio-signal data. [35] Among the many popular Deep Learning architectures leveraged in such tasks, convolutional neural networks (CNNs) are widely used for their ability to learn patterns in structured, multidimensional data (e.g., time-frequency signal representations). However, it is not always clear whether deep learning approaches like CNNs are preferable over more traditional feature-based approaches, especially for short duration bio-signal data. Researchers have demonstrated the accurate inference of movement and gesture activities using CNNs trained on electrophysiological signals acquired from small participant groups. For example, Alias et al.

deployed CNN architectures to classify gait activity of six subjects using surface EMG (sEMG) signals. [36] Each participant of this study performed only six trials of walking according to five different gait modes (walking at slow, normal, and fast speeds on a level surface and walking up and down an incline). Ultimately, their proposed CNN model achieved a gait classification accuracy of 77.95%, a 10.15% increase in accuracy as compared to their traditional, feature-based Support Vector Machine model. Briouza et al. similarly proposed a shallow CNN architecture for highly accurate classification of hand movements when trained on sEMG data from the Ninapro-DB2 dataset (a popular hand-motion classification database consisting of sEMG signals of 49 hand motions from 40 participants). [37,38] A hybrid CNN-LSTM (long short-term memory) network consistently outperformed Random Forest Regression and Support Vector Regression models in intra-session and inter-session evaluations of wrist kinematic estimations for data from six healthy participants. [39] In applying such methodologies to the task of mock-PerfO activity-level classification, 16-class CNN classification models were developed and analyzed. These CNN models were constructed to map 2-dimensional spectrogram representations of the mock-PerfO activity signal segments to a probability distribution over the 16 classes.

In effort to maximize the generalizability of the latent features learned by the trained CNN, data augmentation was employed in effort to maximize the diversity that we see in the training set. Each time a signal segment is read into the training data set, multiple random croppings of this segment are also added to the training set. To an extent, this allowed us to increase the size of our training dataset without collecting additional samples, helping to counter overfitting. To maintain constant length input signals among the mock-PerfO activities that varied in duration, activity segments shorter in duration than the fixed input data duration (30 seconds) were repeated after shifting the segment according to the randomized cropping scheme, while segments longer in duration were truncated to the fixed input data duration via randomized cropping. Data augmentation was not performed for the testing set as it would bias the resulting model performance estimate. Additional techniques applied to reduce model variance included the use of L2 kernel regularization [40] in the convolutional and fully connected model layers and the inclusion of Dropout layers [41] throughout the network. Ultimately, following development and evaluation on training and validation datasets, a shallow CNN, depicted in S1 Fig, was trained, and employed for testing purposes.

## Supporting information

**S1 Table. Features computed from Earable waveforms.** Table shows Feature Name, Unit, Group, and Domain. Each feature was computed for the separated EMG, EOG, and EEG signals separately, unless specifically noted.
(XLSX)

**S2 Table. Variance components design.** Variance components were calculated in this pilot study and recorded in S3 Table for trial repeats, participants, and time of day (time), for each of the 16 mock-PerfO tasks.
(XLSX)

**S3 Table. Spearman correlations and Variance components.** A: ICC values for each of the 16 activities for all Earable parameters. B: %CV values for each of the 16 activities for all Earable parameters. C: Number of tasks analyzed (n), the minimum value (min), maximum value (max), median value (median), mean value (mean), standard deviation of the mean (sd), and

standard error of the mean (se). D: Variance components for each Earable feature for each of the 16 activities as shown in the design in S2 Table.
(XLSX)

**S1 Fig. Mock-PerfO Activity-Level CNN Classifier Architecture.** Architecture diagram of the final CNN implemented for activity classification. A single channel spectrogram computed from the segmented waveform is input to the model at classification time. A probability distribution over each of the 16 activities is output. The activity associated with the highest output likelihood estimate is inferred.
(DOCX)

## Acknowledgments

The authors would like to acknowledge Cynthia PortalCelhay for her contributions to the protocol design, as well as Tong Shen for helpful discussions and contributions to the interpretation of the results of this study.

## Author Contributions

**Conceptualization:** Matthew F. Wipperman, Xuefang Wu, Rinol Alaj, Olivier Harari.

**Data curation:** Matthew F. Wipperman.

**Formal analysis:** Matthew F. Wipperman, Galen Pogoncheff, Yiziying Chen.

**Investigation:** Matthew F. Wipperman, Galen Pogoncheff, Xuefang Wu, Yiziying Chen, Sara C. Hamon.

**Methodology:** Matthew F. Wipperman, Galen Pogoncheff.

**Project administration:** Katrina F. Mateo, Robin R. Deterding, Sara C. Hamon, Tam Vu, Rinol Alaj, Olivier Harari.

**Supervision:** Sara C. Hamon, Rinol Alaj, Olivier Harari.

**Visualization:** Matthew F. Wipperman, Galen Pogoncheff.

**Writing – original draft:** Matthew F. Wipperman, Galen Pogoncheff, Katrina F. Mateo, Oren Levy, Andreja Avbersek.

**Writing – review & editing:** Katrina F. Mateo, Oren Levy, Andreja Avbersek, Sara C. Hamon, Rinol Alaj, Olivier Harari.

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
