## [Decision Letter · Decision Letter 0]

25 Mar 2022

PDIG-D-22-00030

A pilot study of the Earable device to measure facial muscle and eye movement tasks among healthy volunteers

PLOS Digital Health

Dear Dr. Wipperman,

Thank you for submitting your manuscript to PLOS Digital Health. After careful consideration, we feel that it has merit but does not fully meet PLOS Digital Health's publication criteria as it currently stands. Therefore, we invite you to submit a revised version of the manuscript that addresses the points raised during the review process.

Common comments from the reviewers are to provide more clarification on the classification approach (e.g., CNN) and the statistical results (e.g., intraclass correlation, power, internal consistency etc) as well as the generalizability of the data given the small sample size of this pilot study.

We look forward to receiving your revised manuscript.

Kind regards,

Nicole Yee-Key Li-Jessen

Academic Editor

PLOS Digital Health

Journal Requirements:

1. Please amend your detailed Financial Disclosure statement. This is published with the article, therefore should be completed in full sentences and contain the exact wording you wish to be published.

i). State what role the funders took in the study. If the funders had no role in your study, please state: “The funders had no role in study design, data collection and analysis, decision to publish, or preparation of the manuscript.”

ii). If any authors received a salary from any of your funders, please state which authors and which funders.

2. Please update the completed 'Competing Interests' statement. Please declare all competing interests beginning with the statement “I have read the journal's policy and the authors of this manuscript have the following competing interests:”.

3. Please provide separate figure files in .tif or .eps format only and remove any figures embedded in your manuscript file. Please ensure that all files are under our size limit of 20MB.

Please also ensure that all files are under our size limit of 20MB.

For more information about how to convert your figure files please see our guidelines: https://journals.plos.org/digitalhealth/s/figures

4. We notice that your supplementary figures and tables are included in the manuscript file. Please remove them and upload them with the file type 'Supporting Information'. Please ensure that all Supporting Information files are included correctly and that each one has a legend listed in the manuscript after the references list.

Additional Editor Comments (if provided):

Reviewers' comments:

Reviewer's Responses to Questions

**Comments to the Author**

1. Does this manuscript meet PLOS Digital Health’s publication criteria? Is the manuscript technically sound, and do the data support the conclusions? The manuscript must describe methodologically and ethically rigorous research with conclusions that are appropriately drawn based on the data presented.

Reviewer #1: Partly

Reviewer #2: Yes

Reviewer #3: Partly

2. Has the statistical analysis been performed appropriately and rigorously?

Reviewer #1: No

Reviewer #2: Yes

Reviewer #3: No

3. Have the authors made all data underlying the findings in their manuscript fully available (please refer to the Data Availability Statement at the start of the manuscript PDF file)?

Reviewer #1: Yes

Reviewer #2: Yes

Reviewer #3: Yes

4. Is the manuscript presented in an intelligible fashion and written in standard English?

Reviewer #1: Yes

Reviewer #2: Yes

Reviewer #3: Yes

5. Review Comments to the Author

Reviewer #1: This manuscript presents a pilot study that, according to the authors, determines whether the Earable device could

be utilized to objectively measure facial muscle and eye movements. In particular, the paper investigates (1) whether raw EMG, EOG, and EEG Earable device signals could be processed to extract useful features in this context, (2) discuss Earable feature data quality, test re-test reliability, and statistical properties, (3) examine whether derived features could be used to determine the difference between various facial muscle and eye movement activities, and, (4) define what features and feature types are important for such activity level classification.

The manuscript is technically sound, and the presented data tend to support the manuscript conclusions. However, the pilot study only creates limited data, and in that context, it is not clear why the authors have decided to include classification approach that needs lots of data (e.g., CNN models), and compare it with classification approach that can cope with limited data (e.g., Random Forest) only to prove that former will provide better results. 

In addition, due to small data sets, which is by the way point as limitation of the study, it would be good to present some calculations on internal consistency of obtained results prior to the discussion about possible generalization of the results. 

The manuscript describes methodologically sound research with conclusions that are appropriately drawn based on the available data. However, it would be nice to further elaborate on the selection of initial set of activities (PerfOs) with a reference related to their importance for this kind of classification and activity recognition. Further, the pilot study would benefit from integration of “non-healthy” subject(s) and their influence on the proposed quality of extracted features. Once again, this is already pointed out by the authors as limitation of the study.

Reviewer #2: This work studied the utility of earable devices in measuring facial muscle and eye movement tasks among healthy subjects and demonstrated that they can be used for classify mock-PerfO activities. The paper was written clearly and the analysis was performed well. There are a few points the authors might consider to improve the readability of the paper. 

1) Some texts in the Results Section are repeating these in the Methods Section. The authors might consider removing them.

2) The resolution of Figure 1 needs to improved. In particular, the texts are not readable.

3) A layout figure for the sensor and how the sensor was attached would be helpful.

4) The type of intraclass correlation needs to be specified

Reviewer #3: This is an interesting and well-conducted study. Can you explain the choice of the N=10 subjects and how that power was deemed adequate? Similarly, can you describe how the measures taken in healthy volunteers may differ in various patient populations that may benefit from the use of this device?

6. PLOS authors have the option to publish the peer review history of their article (what does this mean?). If published, this will include your full peer review and any attached files.

**Do you want your identity to be public for this peer review?** For information about this choice, including consent withdrawal, please see our Privacy Policy.

Reviewer #1: No

Reviewer #2: No

Reviewer #3: No

---

## [Decision Letter · Decision Letter 1]

9 May 2022

A pilot study of the Earable device to measure facial muscle and eye movement tasks among healthy volunteers

PDIG-D-22-00030R1

Dear Dr Wipperman,

We are pleased to inform you that your manuscript 'A pilot study of the Earable device to measure facial muscle and eye movement tasks among healthy volunteers' has been provisionally accepted for publication in PLOS Digital Health.

Best regards,

Nicole Yee-Key Li-Jessen

Academic Editor

PLOS Digital Health

Reviewer Comments (if any, and for reference):

Reviewer's Responses to Questions

**Comments to the Author**

1. If the authors have adequately addressed your comments raised in a previous round of review and you feel that this manuscript is now acceptable for publication, you may indicate that here to bypass the “Comments to the Author” section, enter your conflict of interest statement in the “Confidential to Editor” section, and submit your "Accept" recommendation.

Reviewer #2: All comments have been addressed

Reviewer #3: All comments have been addressed

2. Does this manuscript meet PLOS Digital Health’s publication criteria? Is the manuscript technically sound, and do the data support the conclusions? The manuscript must describe methodologically and ethically rigorous research with conclusions that are appropriately drawn based on the data presented.

Reviewer #2: Yes

Reviewer #3: Yes

3. Has the statistical analysis been performed appropriately and rigorously?

Reviewer #2: Yes

Reviewer #3: N/A

4. Have the authors made all data underlying the findings in their manuscript fully available (please refer to the Data Availability Statement at the start of the manuscript PDF file)?

Reviewer #2: Yes

Reviewer #3: Yes

5. Is the manuscript presented in an intelligible fashion and written in standard English?

Reviewer #2: Yes

Reviewer #3: Yes

6. Review Comments to the Author

Reviewer #2: (No Response)

Reviewer #3: (No Response)

7. PLOS authors have the option to publish the peer review history of their article (what does this mean?). If published, this will include your full peer review and any attached files.

**Do you want your identity to be public for this peer review?** For information about this choice, including consent withdrawal, please see our Privacy Policy.

Reviewer #2: No

Reviewer #3: No
